# The Assignment Problem and Its Relation to Logistics Problems

Milos Seda 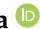

Institute of Automation and Computer Science, Faculty of Mechanical Engineering, Brno University of Technology, Technicka 2896/2, 623 00 Brno, Czech Republic; seda@fme.vutbr.cz

**Abstract:** The assignment problem is a problem that takes many forms in optimization and graph theory, and by changing some of the constraints or interpreting them differently and adding other constraints, it can be converted to routing, distribution, and scheduling problems. Showing such correlations is one of the aims of this paper. For some of the derived problems having exponential time complexity, the question arises of their solvability for larger instances. Instead of the traditional approach based on the use of approximate or stochastic heuristic methods, we focus here on the direct use of mixed integer programming models in the GAMS environment, which is now capable of solving instances much larger than in the past and does not require complex parameter settings or statistical evaluation of the results as in the case of stochastic heuristics because the computational core of software tools, nested in GAMS, is deterministic in nature. The source codes presented may be an aid because this tool is not yet as well known as the MATLAB Optimisation Toolbox. Benchmarks of the permutation flow shop scheduling problem with the informally derived MIP model and the traveling salesman problem are used to present the limits of the software's applicability.

**Keywords:** assignment problem; traveling salesman problem; vehicle routing problem; flow shop scheduling problem; GAMS, genetic algorithm

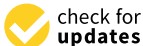

## 1. Introduction

The *Assignment Problem* (abbreviated to AP) [1] and its mathematical model is a problem that is the basis of the field of combinatorial optimization [2,3]. The problem in its basic form has been successfully handled by the discovery of Harold Kuhn, who proved that his method [4], derived from the results of the theoretical work of the Hungarian mathematicians Dénes König and Jenö Egerváry and dubbed the *Hungarian method* in their honor, finds a solution in polynomial time $\mathcal{O}(n^3)$ [5] in an efficient implementation.

However, this does not make the assignment problem less interesting because it has many analogs in bipartite graph matching problems of the graph theory [5,6].

The assignment problem is most extensively addressed in [5,7], where we find theoretical foundations for the existence of perfect matching, implementation details for the Hungarian method, and a number of other related problems such as the *k*-cardinality assignment problem, the semi-assignment problem, the bottleneck assignment problem, the algebraic assignment problem, quadratic assignment problems, and multi-index assignment problems.

These are still variants closely related in meaning to the basic version of the matching problem, although the time complexity may no longer be polynomial, and, in the case of the quadratic matching problem, it is a non-linear problem.

In this paper, however, we focus on similarities of a different kind, namely in the mathematical model, where a completely different relationship to the underlying problem may be found because it allows, often with small modifications, for expressing problems from a different application domain, to switch between linear, mixed integer, and even non-linear problem classes, thus changing its computational complexity and solvability.

Perhaps the most well-known combinatorial optimization problem, the Travelling Salesman Problem [8], is a slight modification of this, with the Vehicle Routing Problem [9]

also stemming from this. On the other hand, from the assignment problem, with other modifications, we can easily move to logistic distribution operations [10], agricultural applications [11], set covering problems [12] with many interesting applications in telecommunications [13], and scheduling problems [14].

Concerning two selected problems of exponential complexity, the Travelling Salesman Problem (TSP) and the Permutation Flow Shop Scheduling Problem (PFSSP) [15], we will also deal with their solvability. We have very good experience in solving sets covering problems of $\mathcal{O}(2^n)$ complexity [13] using GAMS (General Algebraic Modelling System), where it is possible to find the optimal solution in the available time even for instances with matrices of hundreds of rows and thousands of columns, and it has also proven itself in solving the problem of finding the Steiner minimum tree in networks, which also has exponential time complexity.

Since the two problems mentioned above are permutational in nature with the factorial time complexity, they are more challenging than the set covering problems.

For extremely large TSP instances (many hundreds of cities), heuristics must be used, e.g., differential evolution [16], genetic algorithm [17–20], memetic search [21], simulated annealing [22], neural network [23], and improved neighbourhood search algorithms [24,25].

Many stochastic heuristics are inspired by the behaviour of animals in nature, e.g., deer [26], spider monkey [27], hyena [28], wolf [29], cuckoo [30,31], sparrow [32], frog [33], and ant colony [34].

On the other hand, stochastic heuristic methods are not suitable for TSP instances up to 100 cities because they may not find the optimal solution and the convergence time is often unsure, as, e.g., shown in the comparison of different methods in [28].

However, there are also approaches based on deterministic methods such as cutting plane [35], branch and bound [36] and branch and cut [37,38].

According to listings in the GAMS environment, the latter method is in some way incorporated into GAMS and, therefore, it makes sense to explore its limits of applicability. These, together with the GAMS source code, are discussed in detail in Section 6.

Scheduling problems seem far from the assignment problem. But one of them, the PFSSP, shares with the assignment problem a permutational nature in the ordering of jobs, where each job (with its operations) is assigned to exactly one position and each position can contain only one job (with its operations), which corresponds in the assignment problem to the fact that each task is assigned to a single worker and each worker solves only one task. There are additional constraints, and the aim is to minimize the total scheduling time (makespan), but we can still say that the derived PFSSP model is related to the assignment problem.

As in the case of the TSP, heuristic methods are used for large instances of different variants of flow shop scheduling problems, e.g., differential evolution [39], genetic algorithm [40,41], genetic programming [42], memetic algorithm [43], tabu-search [44], harmony search [45], iterated greedy algorithms [46–48], multi-local search [49], hybrid metaheuristics [50,51], reinforcement learning [52], fireworks algorithm [53], and also nature-inspired algorithms, e.g., ant colony optimization [54], firefly particle swarm optimization [55], migrating birds optimization [56] and whale swarm algorithm [57].

However, the exact methods [58,59], linear programming approach [60] and branch and bound [61], are also applicable so we will again focus on the usability of the GAMS tool.

## 2. The Assignment Problem Model

In most common problem formulation, we have $n$ workers who need to be assigned $n$ tasks in such a way that each worker is assigned a single task and each task is solved by a single worker.

For each worker-task pair, we know the time it takes the worker to complete the task. The task is to find an assignment that minimizes the total time to complete all tasks.

Let $c_{ij}$ denote the time taken by the $i$th worker for the $j$th task. The decision variables are binary, $x_{ij} = 1$ if the $i$th worker is assigned the $j$th task, $x_{ij} = 0$ in the opposite case. Then, the problem can be formulated as follows:

$$z = \sum_{i=1}^{n} \sum_{j=1}^{n} c_{ij} x_{ij} \rightarrow \min \tag{1}$$

subject to

$$\sum_{i=1}^{n} x_{ij} = 1, \; j = 1, \ldots, n \tag{2}$$

$$\sum_{j=1}^{n} x_{ij} = 1, \; i = 1, \ldots, n \tag{3}$$

$$x_{ij} \in \{0, 1\}, \; i = 1, \ldots, n, \; j = 1, \ldots, n \tag{4}$$

Equation (2) ensures that each task is assigned to a single worker, and Equation (3) ensures that each worker is assigned a single task.

The assignment problem can also be viewed as a problem of finding a permutation

$$\begin{pmatrix} 1 & 2 & \ldots & n \\ \pi_1 & \pi_2 & \ldots & \pi_n \end{pmatrix},$$

where $i$th worker is assigned to task $\pi_i$ and

$$z = \sum_{i=1}^{n} c_{i\pi_i} \rightarrow \min$$

Since the number of different permutations of $n$ elements is $n!$, it is not possible to find the optimal solution for large instances in the available time by enumerating all possibilities. However, due to the Hungarian method mentioned above, we no longer use this approach.

## 3. Routing Problems

With a different interpretation of the variables and a possible extension of the constraints, the assignment problem changes into a series of other problems. In this section, we consider two routing problems.

### 3.1. Travelling Salesman Problem

The *Travelling Salesman Problem* (TSP) [8,62] is mathematically similar to the assignment problem model, differing only in one additional constraint, but the meaning of the decision variables $x_{ij} = 0$ is different. It is formulated as follows: Given $n$ cities and distances among them, the objective is to find a round trip through all cities with a minimum length (alternatively, with a minimum total transportation cost).

Since the starting city 1 is fixed, the number of routes is given by the permutations of cities 2, 3, ..., $n$, and is therefore equal to $(n-1)!$. If there are no one-way segments anywhere in the transportation between cities, routes in reverse order of cities do not affect the length, and then we can reduce the number of routes to $(n-1)!/2$, but still the time complexity of exploring all routes is $\mathcal{O}(n!)$.

If we denote by $c_{ij}$ the distance between cities $i$ and $j$ (alternatively, the price of transportation between cities $i$ and $j$), $x_{ij}$ a binary decision variable that takes the value 1 when city $j$ on the route immediately follows city $i$, otherwise it takes the value 0, $\delta_i$ is the order of city $i$ on the route, then the Travelling Salesman Problem can be formulated as follows:

$$z = \sum_{i=1}^{n} \sum_{j=1}^{n} c_{ij} x_{ij} \rightarrow \min \tag{5}$$

subject to

$$\sum_{i=1}^{n} x_{ij} = 1, \ j = 1, \ldots, n \tag{6}$$

$$\sum_{j=1}^{n} x_{ij} = 1, \ i = 1, \ldots, n \tag{7}$$

$$\delta_i - \delta_j + n x_{ij} \le n - 1, \ i \ne j, \ i = 2, \ldots, n, \ j = 2, \ldots, n \tag{8}$$

$$x_{ij} \in \{0, 1\}, \ i = 1, \ldots, n, \ j = 1, \ldots, n \tag{9}$$

Constraints (6) and (7) ensure that each city (vertex of the graph) is traversed exactly once (entered and left exactly once); the system of subtour elimination constraints (8), referred to in the English literature by Miller, Tucker, and Zemlin as *MTZ constraints*, prevents the formation of subtours, as we will show below in Theorem 1.

Equation (8) follows from the following reasoning:
(i) If $x_{ij} = 1$, then $j$ is the immediate successor of $i$, and if $\delta_i = t$, then $\delta_j = t + 1$. Hence, $\delta_i - \delta_j + n x_{ij} = t - (t + 1) + n = n - 1$.
(ii) If $x_{ij} = 0$, then $\delta_i - \delta_j + n x_{ij} = \delta_i - \delta_j$ and this difference in the order of the cities in the route for $i \ne j$ can be at most equal to $n - 1$.

From (i) and (ii), a common conclusion $\delta_i - \delta_j + n x_{ij} \le n - 1$ already follows, which, for all combinations of feasible values of $i$ and $j$, is expressed by inequality (8).

Without the constraint (8), constraints (6) and (7) are satisfied by splitting the route into several subtours, e.g., for 15 vertices, the two conditions mentioned above are satisfied by the subtours $1 - 3 - 7 - 9 - 12 - 1$, $2 - 4 - 10 - 11 - 13 - 15 - 2$ and $5 - 6 - 8 - 14 - 5$.

**Theorem 1.** (Miller, Tucker, Zemlin) *The variables $x_{ij} \in \{0, 1\}$, $i = 1, \ldots, n$, $j = 1, \ldots, n$ satisfying constraints (6) and (7) form a Hamiltonian circle if and only if the subtour elimination constraints (8) are satisfied.*

**Proof.** Suppose that $x_{ij}$ satisfies the subtour elimination constraints but does not form a Hamiltonian circle. Then $x_{ij}$ due to (6) and (7) form at least two subtours, one containing the initial vertex 1 and another without it. Let $S$ be a subtour that does not contain vertex 1 and let $E(S)$ be the set of edges in $S$. Summing the conditions over the edges of $E(S)$ we get:

$$\sum_{(i,j) \in E(S)} (\delta_i - \delta_j + n x_{ij}) \le (n - 1)|E(S)|,$$

since the values of $\delta_i$ and $\delta_j$ eliminate each other in this subtour, we get

$$n|E(S)| \le (n - 1)|E(S)|,$$

which is a contradiction.

Assume now that $x_{ij}$ forms a Hamiltonian circle. If 1 is the initial vertex of this circle, and for each vertex $i \ne 1$, $\delta_i = k$, if $i$ is the $k$th vertex of the Hamiltonian circle, then it is clear that the conditions (8) are satisfied.　□

### 3.2. Vehicle Routing Problem

A generalization of the Travelling Salesman Problem is the *Vehicle Routing Problem* (VRP) [9,63–66] where a desired quantity of goods needs to be delivered from a central depot to customers by vehicles of a certain capacity.

We are looking for closed routes of individual vehicles that start and end at the depot, each customer is served exactly once by exactly one vehicle, the requirements of all customers are met and the total transport costs are minimal.

Consider the following notation:
$n$ ... number of customers
$0$ ... depot (start and end of each vehicle's route)

$K$ ... number of (identical) vehicles

$d_j \geq 0$ ... request of the $j$th customer (for depot $d_0 = 0$)

$Q$ ... vehicle capacity $(KQ \geq \sum_{j=1}^{n} d_j)$

$c_{ij}$ ... the cost of transport from $i$ to $j$ $(c_{ii} = 0)$

$x_{ij}$ ... binary decision variable equal to 1 if $j$ is immediately followed by $i$ on the route, $x_{ij} = 0$ otherwise

$\delta_i$ ... the load left in the vehicle after visiting customer $i$.

$$z = \sum_{i=0}^{n} \sum_{j=0}^{n} c_{ij} x_{ij} \rightarrow \min \tag{10}$$

subject to

$$\sum_{i=0}^{n} x_{ij} = 1, \ j = 1, \ldots, n \tag{11}$$

$$\sum_{j=0}^{n} x_{ij} = 1, \ i = 1, \ldots, n \tag{12}$$

$$\sum_{i=1}^{n} x_{i0} = K \tag{13}$$

$$\sum_{j=1}^{n} x_{0j} = K \tag{14}$$

$$0 \leq \delta_i \leq Q - d_i, \ i = 1, \ldots, n \tag{15}$$

$$\delta_i - \delta_j + Q x_{ij} \leq Q - d_j, \ i \neq j, \ i = 1, \ldots, n, \ j = 1, \ldots, n, \ \text{such that } d_i + d_j \leq Q \tag{16}$$

$$x_{ij} \in \{0,1\}, \ i = 0, \ldots, n, \ j = 0, \ldots, n \tag{17}$$

In the model, (11) and (12) ensure that exactly one vehicle arrives at each customer (11) and exactly one vehicle leaves it (12). Equations (13) and (14) ensure that all $K$ vehicles return to the depot (13) and all $K$ vehicles leave the depot (14).

Equation (16) is analogous to the MTZ constraints in the Travelling Salesman Problem preventing the formation of partial circuits and at the same time ensuring that the requirements of customers $i$ and $j$ can be met when traveling from $i$ to $j$ [67].

The Vehicle Routing Problem has many other specific formulations, e.g., there may be a larger number of depots available, and customers are only ready to receive delivery of goods at certain time intervals. For more details, see the sources listed at the beginning of this section.

## 4. Distribution Problems

Distribution problems have many different formulations, first, we consider the classical Hitchcock's *Transportation/Transshipment Problem* with $m$ suppliers (sources, warehouses) and $n$ customers (consumers), where we assume the transportation of a single type of material (goods) with an objective to minimize the total cost of transporting the material [68].

Assume the following notation:

$a_i, \ i = 1, \ldots, m$ ... capacity (stocks) of suppliers,

$b_j, \ j = 1, \ldots, n$ ... customer requirements,

$c_{ij}, \ i = 1, \ldots, m, \ j = 1, \ldots, n$ ... the matrix of rates for the transport of a unit quantity between the $i$th supplier and the $j$th customer,

$x_{ij}, \ i = 1, \ldots, m, \ j = 1, \ldots, n$ ...the sought quantity transported between the $i$th supplier and the $j$th customer.

If total stocks are equal to total requirements, this means:

$$\sum_{i=1}^{m} a_i = \sum_{j=1}^{n} b_j, \tag{18}$$

we are talking about a *balanced distribution problem*, where all stocks are exhausted and all demands are met, and the following mathematical model corresponds to this:

$$z = \sum_{i=1}^{m} \sum_{j=1}^{n} c_{ij} x_{ij} \rightarrow \min \tag{19}$$

subject to

$$\sum_{j=1}^{n} x_{ij} = a_i, \ i = 1, \ldots, m \tag{20}$$

$$\sum_{i=1}^{m} x_{ij} = b_j, \ j = 1, \ldots, n \tag{21}$$

$$x_{ij} \geq 0, \ i = 1, \ldots, m, \ j = 1, \ldots, n \tag{22}$$

Equation (20) corresponds to the stock drawdown, and Equation (21) expresses the fulfillment of requirements.

Obviously, the assignment problem is a special case of the balanced transportation problem, where:

$$m = n$$

$$a_i = 1, \ i = 1, \ldots, m$$

$$b_j = 1, \ j = 1, \ldots, n$$

$$x_{ij} \in \{0, 1\}, \ i = 1, \ldots, m, \ j = 1, \ldots, n$$

However, the balanced case is rare in practice, usually, total stocks exceed total requirements, i.e.,

$$\sum_{i=1}^{m} a_i > \sum_{j=1}^{n} b_j \tag{23}$$

In this case, all requirements can be met, but not every stock will be used up. The model then changes as follows:

$$z = \sum_{i=1}^{m} \sum_{j=1}^{n} c_{ij} x_{ij} \rightarrow \min \tag{24}$$

subject to

$$\sum_{j=1}^{n} x_{ij} \leq a_i, \ i = 1, \ldots, m \tag{25}$$

$$\sum_{i=1}^{m} x_{ij} = b_j, \ j = 1, \ldots, n \tag{26}$$

$$x_{ij} \geq 0, \ i = 1, \ldots, m, \ j = 1, \ldots, n \tag{27}$$

In the case of material shortages, the opposite situation may occur, where the total stock is insufficient for the total requirements, i.e.,

$$\sum_{i=1}^{m} a_i < \sum_{j=1}^{n} b_j \tag{28}$$

This means that stocks are used up but not all requirements can be met. The model must then be modified as follows:

$$z = \sum_{i=1}^{m} \sum_{j=1}^{n} c_{ij} x_{ij} \rightarrow \min \tag{29}$$

subject to

$$\sum_{j=1}^{n} x_{ij} = a_i, \ i = 1, \ldots, m \tag{30}$$

$$\sum_{i=1}^{m} x_{ij} \leq b_j, \ j = 1, \ldots, n \tag{31}$$

$$x_{ij} \geq 0, \ i = 1, \ldots, m, \ j = 1, \ldots, n \tag{32}$$

### 4.1. Container Transportation Problem

The Container Transportation Problem is a special case of Hitchcock's transportation problem, where we assume that materials from suppliers to customers are transported only in containers of a certain capacity. Instead of rates per unit of material transported, there are prices per container transported, being fixed even if the container is not completely full.

From the previous three possibilities for the sum of all stocks and the sum of all requirement relations, the case of the stocks being sufficient to meet all the requirements is given here.

Assume that $K$ is the capacity of the container and $y_{ij}$ gives the number of containers needed for the quantity of material $x_{ij}$. Obviously, $y_{ij}$ must be integers, the last container to reach the quantity $x_{ij}$ need not be full.

Then, the container transportation problem for all requirements met can be formulated as the following model:

$$z = \sum_{i=1}^{m} \sum_{j=1}^{n} c_{ij} y_{ij} \rightarrow \min \tag{33}$$

subject to

$$\sum_{j=1}^{n} x_{ij} \leq a_i, \ i = 1, \ldots, m \tag{34}$$

$$\sum_{i=1}^{m} x_{ij} = b_j, \ j = 1, \ldots, n \tag{35}$$

$$x_{ij} \leq K y_{ij}, \ i = 1, \ldots, m, \ j = 1, \ldots, n \tag{36}$$

$$y_{ij} \in \mathbb{Z}_+, \ i = 1, \ldots, m, \ j = 1, \ldots, n \tag{37}$$

$$x_{ij} \geq 0, \ i = 1, \ldots, m, \ j = 1, \ldots, n \tag{38}$$

### 4.2. Allocation Problem

For the transportation problem described in Section 4, it was possible to provide the required quantity by composing partial quantities from different suppliers (from different warehouses) when fulfilling the requirements.

However, in the *Allocation Problem*, it is required that the required quantity is provided from a single location so the mathematical model of the transportation problem has to be modified by [10] to account for this condition. With the same notation used for the symbols, the meaning of the decision variables $x_{ij}$ is now different. They only assume binary values and $x_{ij} = 1$ if the quantity $b_j$ required by the $j$th customer is sourced from the $i$th supplier, if not, $x_{ij} = 0$.

If more than one customer receives the required quantity from the same supplier, the sum of their requirements must not exceed the capacity of that supplier (stock).

The model of the allocation problem with these conditions then takes the following form:

$$z = \sum_{i=1}^{m} \sum_{j=1}^{n} c_{ij} x_{ij} \rightarrow \min \tag{39}$$

subject to

$$\sum_{i=1}^{m} x_{ij} = 1, \ j = 1, \ldots, n \tag{40}$$

$$\sum_{j=1}^{n} b_j x_{ij} \leq a_i, \ i = 1, \ldots, m \tag{41}$$

$$x_{ij} \in \{0,1\}, \ i = 1, \ldots, m, \ j = 1, \ldots, n \tag{42}$$

### 4.3. Location Problem

The *Location Problem* is an extension of the allocation problem [12]. For clarity, let us first summarize all the symbols used.

Consider $m$ locations with capacities $a_i$, $i = 1, \ldots, m$ that can be used to operate warehouses supplying $n$ customers with demands $b_j$, $j = 1, \ldots, n$. The operation of the warehouse at the $i$th location requires a cost $f_i$, $i = 1, \ldots, m$ for the given period. Let $c_{ij}$, $i = 1, \ldots, m$, $j = 1, \ldots, n$ be the cost of the $j$th customer being assigned to get the required quantity from the $i$th location.

The aim is to decide in which locations to operate the warehouses and to find the assignment of customers to the operated warehouses so that the value of the total cost of operating the system is minimal. Like in the allocation problem, we assume that the demands of each consumer must be covered from a single warehouse.

Therefore, the meaning of the binary decision variables $x_{ij}$ is analogous to the allocation problem, $x_{ij} = 1$, if the quantity $b_j$ required by the $j$th customer is provided from the warehouse at the $i$th location, if not, $x_{ij} = 0$.

In addition, there are other binary decision variables $y_i$, $i = 1, \ldots, m$, where $y_i = 1$ means that the warehouse at the $i$th location will be operated and, if $y_i = 0$, it will not be operated there.

The model of the location problem with these conditions has the following form:

$$z = \sum_{i=1}^{m} \sum_{j=1}^{n} c_{ij} x_{ij} + \sum_{i=1}^{m} f_i y_i \rightarrow \min \tag{43}$$

subject to

$$\sum_{i=1}^{m} x_{ij} = 1, \ j = 1, \ldots, n \tag{44}$$

$$x_{ij} \leq y_i, \ i = 1, \ldots, m, \ j = 1, \ldots, n \tag{45}$$

$$\sum_{j=1}^{n} b_j x_{ij} \leq a_i, \ i = 1, \ldots, m \tag{46}$$

$$x_{ij} \in \{0,1\}, \ i = 1, \ldots, m, \ j = 1, \ldots, n \tag{47}$$

$$y_i \in \{0,1\}, \ i = 1, \ldots, m \tag{48}$$

As in the allocation problem, the condition (44) means that each customer takes the entire requested quantity from a single location, the condition (46) monitors the non-overstocking of individual locations by customers receiving the requested quantity from the same location.

Let us have a look at condition (45). The left and right sides are binary variables with the inequality satisfied for the combinations $0 \leq 0$, $0 \leq 1$ and $1 \leq 1$, but not for the

combination $1 \leq 0$. This ensures that no customer can get anything from a location where the warehouse will not be operated.

It is clear that, for all combinations of indices $i$ and $j$, (45) represents a system of $mn$ conditions. Expressing this for the values of the indices $j$, we get:

$$x_{i1} \leq y_i, \ i = 1, \ldots, m$$

$$x_{i2} \leq y_i, \ i = 1, \ldots, m$$

$$\vdots$$

$$x_{in} \leq y_i, \ i = 1, \ldots, m$$

Summing up the previous inequalities, we get:

$$x_{i1} + x_{i2} + \cdots + x_{in} \leq y_i + y_i + \cdots + y_i \ i = 1, \ldots, m,$$

and hence

$$\sum_{j=1}^{n} x_{ij} \leq n y_i, \ i = 1, \ldots, m \tag{49}$$

Equation (49) is equivalent to (45), but is simpler because it represents only $m$ conditions rather than the $mn$ conditions in the original expression (45).

### 4.4. Capacitated Network Area Coverage

Let us consider two finite sets $I$ and $J$, where $I$ is the set of service centers $1, 2, \ldots, m$, and $J$ is the set of customer locations $1, 2, \ldots, n$.

Further, $a_{ij} = 1$ means that customer location $j$ is in a reachable distance to service center $i$, $a_{ij} = 0$ means that it does not satisfy it, and $w_i$ expresses the weights of service centers (since it is the minimization problem, the greater the weights are, the smaller the coefficient must be).

Similarly, $x_i = 1$ means that service centre $i$ is selected, while $x_i = 0$ means that it is not selected.

Finally, $c_i$, $i \in I$—capacity of service centre $i$, $b_j$, $j \in J$—demand of customer location $j$, $y_{ij} \in \{0, 1\}$—customer from location $j$ is assigned or is not assigned to service centre $i$.

In [13], we derived the following model for a capacitated network area coverage:

$$z = \sum_{i \in I} w_i x_i \rightarrow \min \tag{50}$$

subject to

$$\forall j \in J : \sum_{i \in I} a_{ij} x_i \geq 1 \tag{51}$$

$$\forall j \in J : \sum_{i \in I} a_{ij} y_{ij} = 1 \tag{52}$$

$$\forall i \in I : c_i x_i \geq \sum_{j \in J} a_{ij} y_{ij} b_j \tag{53}$$

$$\forall i \in I : \sum_{j \in J} y_{ij} \leq n x_i \tag{54}$$

$$\forall i \in I : x_i \in \{0, 1\} \tag{55}$$

$$(\forall i \in I)(\forall j \in J) : y_{ij} \in \{0, 1\}. \tag{56}$$

A necessary precondition for finding a solution is that the sum of all capacities is sufficient to cover all demands, i.e., $\sum_{i=1}^{m} c_i \geq \sum_{j \in J} b_j$, with each customer having a reachable distance to at least one center, i.e., $\forall j \in J : \sum_{i \in I} a_{ij} > 0$.

In [13], we then modified the previous model for the domain of telecommunication signals considering signal interference and its nonlinear version linearized as follows:

$$z = \left( \sum_{i \in I} w_i x_i \right) / \sum_{i \in I} w_i - \left( \sum_{i \in I} \sum_{j \in I} d_{ij} h_{ij} \right) / \left( \sum_{i \in I} \sum_{j \in I} d_{ij} \right) \to \min \tag{57}$$

subject to

$$(\forall i \in I)(\forall j \in I): \ h_{ij} \leq x_i \tag{58}$$

$$(\forall i \in I)(\forall j \in I): \ h_{ij} \leq x_j \tag{59}$$

$$(\forall i \in I)(\forall j \in I): \ h_{ij} \geq (x_i + x_j - 1) \tag{60}$$

$$(\forall i \in I)(\forall j \in I): \ h_{ij} \in \{0,1\} \tag{61}$$

$$\forall j \in J: \ \sum_{i \in I} a_{ij} x_i \geq 1 \tag{62}$$

$$\forall j \in J: \ \sum_{i \in I} a_{ij} y_{ij} = 1 \tag{63}$$

$$\forall i \in I: \ c_i x_i \geq \sum_{j \in J} a_{ij} y_{ij} b_j \tag{64}$$

$$\forall i \in I: \ \sum_{j \in J} y_{ij} \leq n x_i \tag{65}$$

$$(\forall i \in I)(\forall j \in I)(i \neq j): \ d_{ij} \geq (x_i + x_j - 1)d_{\min} \tag{66}$$

$$\forall i \in I: \ x_i \in \{0,1\} \tag{67}$$

$$(\forall i \in I)(\forall j \in J): \ y_{ij} \in \{0,1\}. \tag{68}$$

Another possible modification of the model is to meet the demand by composing parts of the capacities of several centers, but with a fragmentation not lower than a certain threshold. This new approach will be presented in detail in a separate paper.

*4.5. Transportation Problem with Supply from Primary Source*

Consider now a transportation network where, in addition to locations with warehouse stocks and customer requirements, there will also be a primary source, which can represent the location of the transported commodity or a global warehouse, and customers can be supplied both from local warehouses and directly from the primary source.

Assume the constraints and denotations from the location problem and two types of transportation equipment, one with a larger capacity $k_1$ from the primary source to local warehouses and a cost $n_1$ per 1 km of travel, and the other with a smaller capacity $k_2$ to customers and a cost $n_2$ per 1 km. Denoting the distance from the primary source to the $i$th local storage by $e_i$, and the distance from the primary source to the $j$th customer by $g_j$, we add binary decision variables $z_j$ to indicate whether the $j$th customer receives the desired quantity directly from the primary source (in the positive case $z_j = 1$, otherwise $z_j = 0$), the model with primary source and transportation technique information has the following form:

$$z = \sum_{j=1}^{n} \frac{b_j}{k_2} g_j n_2 z_j + \sum_{i=1}^{m} \sum_{j=1}^{n} \left( \frac{b_j}{k_1} e_i n_1 + \frac{b_j}{k_2} d_{ij} n_2 \right) x_{ij} + \sum_{i=1}^{m} f_i y_i \to \min \tag{69}$$

subject to

$$z_j + \sum_{i=1}^{m} x_{ij} = 1, \ j = 1, \dots, n \tag{70}$$

$$\sum_{j=1}^{n} x_{ij} \leq n y_i, \; i = 1, \ldots, m \tag{71}$$

$$\sum_{j=1}^{n} b_j x_{ij} \leq a_i, \; i = 1, \ldots, m \tag{72}$$

$$x_{ij} \in \{0,1\}, \; i = 1, \ldots, m, \; j = 1, \ldots, n \tag{73}$$

$$y_i \in \{0,1\}, \; i = 1, \ldots, m \tag{74}$$

$$z_j \in \{0,1\}, \; j = 1, \ldots, n \tag{75}$$

The fractions in the objective function according to (69) indicate how many times the distance must be traveled for the customer to receive the required quantity. Since the fractions may have non-integer values, they must be rounded up to integers, the corresponding capacity may not be fully used for the last trip. The expression with the first summation in the objective function corresponds to the total cost of moving material from the primary source directly to customers, and the expression with the double summation in the objective function corresponds to the total cost of moving material from the primary source to local warehouses and from there on to the customers.

If, instead of the conditions of the location problem, the simpler conditions of the allocation problem were assumed (i.e., dropping the decision as to whether or not to use a location for storage), the previous model would be simplified, the conditions (71) and (74) would be dropped and $y_i$ would be omitted in the last term of the objective function (i.e., the fixed costs of all locations would be included).

*4.6. Crop Problem*

In plant production, an important task is to find a method of sowing the land with agricultural crops (cultures) in such a way that, given the expected yield of crops on the land and the profit from the sale of individual crops, the total profit is maximised.

Assume the following notation:
$p_i$, $i = 1, \ldots, m$ ... grounds,
$r_i$, $j = 1, \ldots, m$ ... area of grounds (plays the role of available capacities),
$k_j$, $j = 1, \ldots, n$ ... agricultural crops (cultures),
$c_{ij}$, $i = 1, \ldots, m$, $j = 1, \ldots, n$ ... profit from 1 ha of ground $p_i$, sown with culture $k_j$
$x_{ij}$, $i = 1, \ldots, m$, $j = 1, \ldots, n$ ...number of hectares of ground $p_i$ sown with crop $k_j$.

The mathematical model of the Crop Problem is similar to the basic version of the transportation problem with unbalanced capacities (the ground areas may not be fully used), but it lacks a set of constraints corresponding to the fulfillment of the requirements with the difference that the problem being a maximization one. It takes the following form:

$$z = \sum_{i=1}^{m} \sum_{j=1}^{n} c_{ij} x_{ij} \rightarrow \max \tag{76}$$

subject to

$$\sum_{j=1}^{n} x_{ij} \leq r_i, \; i = 1, \ldots, m \tag{77}$$

$$x_{ij} \geq 0, \; i = 1, \ldots, m, \; j = 1, \ldots, n \tag{78}$$

Equation (77) expresses the use of grounds, which corresponds to the drawdown of supplier stocks in the distribution problem.

If we required each crop $j$ to be sown on some minimum area $d_j$, then the problem would become an example of the maximization version of the *generalized distribution problem* and the model would be modified as follows:

$$z = \sum_{i=1}^{m} \sum_{j=1}^{n} c_{ij} x_{ij} \rightarrow \max \tag{79}$$

subject to

$$\sum_{j=1}^{n} x_{ij} \leq r_i, \ i = 1, \dots, m \tag{80}$$

$$\sum_{i=1}^{m} x_{ij} \geq d_j, \ j = 1, \dots, n \tag{81}$$

$$x_{ij} \geq 0, \ i = 1, \dots, m, \ j = 1, \dots, n \tag{82}$$

## 5. Scheduling Problems

Scheduling problems are numerous and varied. They arise in diverse areas such as flexible manufacturing systems, production planning, computer design, logistics, timetabling, communication, etc. [14].

Here we focus on one of them, the Permutation Flow Shop Scheduling Problem (PFSSP), which, like the Assignment problem and the Traveling Salesman Problem, is permutational in nature.

It can be briefly described as follows: There are a set of $m$ machines (processors) and a set of $n$ jobs. Each job comprises a set of $m$ operations which must be done on different machines. All jobs have the same processing operation order when passing through the machines. There are no precedence constraints among operations of different jobs. Operations cannot be interrupted, and each machine can process only one operation at a time. The problem is to find the job sequences on the machines which minimizes the makespan (i.e., the maximum completion times of all operations).

*Mathematical Model of PFSSP*

Consider three finite sets $J, M, O$ where $J$ is a set of jobs $1, \dots, n$, $M$ is a set of machines $1, \dots, m$, and $O$ is a set of operations $1, \dots, m$.

Denote:

$J_i$ ... the $i$th job in the permutation of jobs
$p_{ik}$ ... processing time of the job $J_i \in J$ on machine $k$

$(\forall i \in J)(\forall k \in M) : \quad v_{ik} \ = \ $ waiting time (idle time) on machine $k$ before starting job $J_i$

$(\forall i \in J)(\forall k \in M) : \quad w_{ik} \ = \ $ waiting time (idle time) of job $J_i$ after finishing processing on machine $k$ while waiting for machine $k+1$ to become available

Define the following decision variables:

$$\forall i, j \in J : \ x_{ij} = \begin{cases} 1, & \text{if job } j \text{ is assigned to the } i\text{th position in the permutation } (J_i = j) \\ 0, & \text{otherwise} \end{cases} \tag{83}$$

Figure 1 illustrates the use of the variables $v_{ik}$ and $w_{ik}$ on an example with 5 jobs and 3 machines.

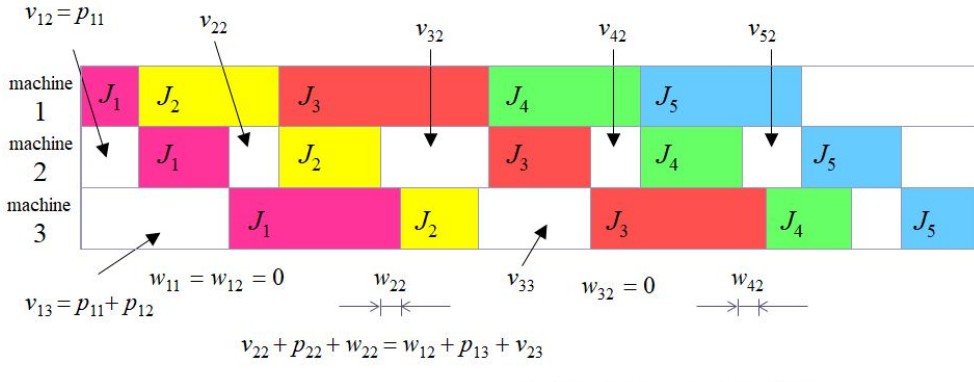

**Figure 1.** Meaning of the variables $v_{ik}$ and $w_{ik}$.

From Figure 1, we can draw some more general conclusions:

- The first task in a permutation can always continue the next operation on the next machine without delay because it does not wait for the completion of any other operation.
- It follows from the previous conclusion that waiting times to start the operation of the first task in the permutation on the second and subsequent machines are given by the sum of the durations of the operations of that task on the previous machines.
- Equalities of 3 addition terms in Figure 1 can be generalized into a Gantt chart between all pairs of neighboring machines.
- The duration of the entire schedule (makespan) is given by the sum of the waiting times for the start of operations on the last machine and the duration of these operations.

All verbal conclusions are expressed formally by the following system of equations:

$$\forall i \in J : \sum_{j=1}^{n} x_{ij} = 1 \tag{84}$$

$$\forall j \in J : \sum_{i=1}^{n} x_{ij} = 1 \tag{85}$$

$$\forall k \in M - \{m\} : w_{1k} = 0 \tag{86}$$

$$\forall k \in M - \{1\} : v_{1k} = \sum_{r=1}^{k-1} \sum_{i=1}^{n} p_{ir} x_{1i} \tag{87}$$

$$(\forall i \in J - \{n\}) (\forall k \in M - \{m\}) :$$

$$v_{i+1,k} + \sum_{j=1}^{n} p_{jk} x_{i+1,j} + w_{i+1,k} = w_{ik} + \sum_{j=1}^{n} p_{j,k+1} x_{ij} + v_{i+1,k+1} \tag{88}$$

$$C_{\max} = \sum_{i=1}^{n} \left( v_{im} + \sum_{j=1}^{n} p_{jm} x_{ij} \right) \tag{89}$$

## 6. Computational Results

From the above problems, we select two, TSP and PFSSP, that are NP-hard in the decision versions [69,70].

To give an idea of the cardinality of the search space of these permutation problems, we present a few factorials as follows:

10! = 3628800 ≈ $3.6 \times 10^6$

20! = 2432902008176640000 ≈ $2.4 \times 10^{18}$

30! = 265252859812191058636308480000000 ≈ $2.6 \times 10^{32}$

40! = 815915283247897734345611269596115894272000000000 ≈ $8.1 \times 10^{47}$

50! ≈ $3 \times 10^{64}$

...

$100! \approx 9.3 \times 10^{157}$

The traditional approaches to such problems are based on computations using heuristic methods [71,72] for large instances such as genetic algorithms, simulated annealing, tabu search, differential evolution [73], firefly algorithm, particle swarm optimization, and ant colony optimization. Then, statistical tests are applied to examine at a certain significance level (e.g., $\alpha = 0.05$), to what extent the mean value of the results obtained by different methods and different settings of their parameters at a larger number of runs is the same or different (and, therefore, one of the methods gives better results). For the *t*-test, we assume that the sets of values have a normal distribution. However, this assumption may be false, and then one of the non-parametric tests, such as the Wilcoxon test, must be used.

Since, given the validity of the No Free Lunch Theorem [74,75], one should not expect a general conclusion that any of the heuristics for each problem instance gives better results than other heuristics.

In this paper, we do not explore heuristics using instead a mixed integer programming model with software tools built as solvers in the GAMS environment [76,77] to find an exact solution by deterministic computation.

Statistical evaluations are, therefore, meaningless here. What can be said, however, is that the power of this software today is considerably greater than it was 20 years ago, when, in our experience, for a problem with a complexity of $\mathcal{O}(20!)$, the system ended up with a runtime error and the message "`insufficient space to update U-factor` ...". The performance of GAMS has been steadily increasing over the years, although the source code of the solvers is not freely available, from [78] it can at least be seen that it includes, among others, CPLEX, GUROBI, Lindo, and the results of the work of academic departments of Princeton University, Stanford University, and Zuse Institute Berlin. Today, GAMS calculates the exact solution for PFSSP with 20 jobs on a laptop with Intel(R) Core(TM) i5-10210U CPU @ 1.60 GHz 2.11 GHz processor, 8 GB operational memory and 64-bit operating system in less than 3 min, as shown in the following subsection.

Of course, with a computer of better technical parameters for the same time limit we get results for larger instances of the problem, but it seems to be better to use a heuristic beyond this boundary, e.g., our GA 'war elimination' modification [79].

Since PFSSP has a more complex model than TSP, we start with it and include its complete GAMS code.

### 6.1. PFSSP Computational Results

For PFSSP with 10 jobs, 6 machines, processing times from the TABLE section (it corresponds to the first benchmark in Table 1) and the model given by Equations (84)–(89), the program code in GAMS can be, e.g., as follows:

```
* Permutation flow shop scheduling problem
$TITLE permutation flow shop scheduling problem
$OFFSYMXREF
$OFFUELLIST
$OFFUELXREF

OPTION ITERLIM=200,000
* ITERLIM number of iterations
OPTION OPTCR=0.00001
*OPTION OPTCR=0.001
* OPTCR stopping in MIP in case the best solution is within the limits
* 100*OPTCR% of the global~extreme

* section defining indexes
SETS
  I jobs /1*10/
  K machines /1*6/;
```



```
   ALIAS(I,J);
* J - position of the job in the permutation
   ALIAS(K,R);

* input data section
PARAMETERS
   N,M;
   N=CARD(I);
   M=CARD(K);
TABLE P(I,K)
          1     2     3     4     5     6
    1    333   991   996   123   145   234
    2    333   111   663   456   785   532
    3    252   222   222   789   214   586
    4    222   204   114   876   752   532
    5    255   477   123   543   143   142
    6    555   566   456   210   698   573
    7    558   899   789   124   532    12
    8    888   965   876   537   145    14
    9    889   588   543   854   247   527
   10    999   889   210   632   451   856;

*variables section (decision variables and objective function)
VARIABLES
   X(I,J) is 1 if job j is assigned to position i in the permutation, 0, otherwise
   V(I,K) waiting time on machine k before the start of job i in the permutation
   W(I,K) waiting time of job i in the permutation after finishing processing
* on machine k, while machine k+1 becomes free
   Cmax total processing time for all tasks (makespan);
BINARY VARIABLE X;
NONNEGATIVE VARIABLE V;
NONNEGATIVE VARIABLE W;

*section describing the system of (in)equalities
EQUATIONS
   EQ1(I)
   EQ2(J)
   EQ3(K)
   EQ4(K)
   EQ5(I,K)
   OBJFCE(K);
   EQ1(I) .. SUM(J,X(I,J)) =E= 1;
   EQ2(J) .. SUM(I,X(I,J)) =E= 1;
   EQ3(K)$(ORD(K) LE (M-1)) .. W(''1'',K) =E= 0;
   EQ4(K)$(ORD(K) GE 2)
      .. V(''1'',K) =E= SUM((R,I)$(ORD(R) LE (ORD(K)-1)),P(I,R)*X(''1'',I));
   EQ5(I,K)$((ORD(I) LE (N-1)) AND (ORD(K) LE (M-1)))
    .. V(I+1,K)+SUM(J,P(J,K)*X(I+1,J))+W(I+1,K) =E=
       W(I,K)+SUM(J,P(J,K+1)*X(I,J))+V(I+1,K+1);
   OBJFCE(K)$(ORD(K) EQ M) .. Cmax =E= SUM(I,V(I,K)+SUM(J,P(J,K)*X(I,J)));

*description of the model, running the solver, and displaying the results
MODEL FLOWSHOP /ALL/;
SOLVE FLOWSHOP USING MIP MINIMIZING Cmax;
DISPLAY X.L, V.L, W.L, Cmax.L;
```

The ability to compute optimal solutions was checked using standard benchmarks from OR-Library (OR = Operations Research) accessible at Brunel University London [80], originally described in [81]. The computational results are summarised in Table 1. For

instances with 30 or more jobs, GAMS does not find the optimal solution in the 1000 s time limit, but only a "close" approximation, which, however, differs by less than 10% even for the last instance $75 \times 20$, where the optimal value is unknown due to the huge size of the search space and is only estimated by the interval. For such cases, we at least suggest a solution method using the genetic algorithm [82].

**Table 1.** GAMS computational results ($10 \times 6$ corresponds to 10 jobs and 6 machines, etc.; t-l-e = time limit exceeded).

| Benchmark | Result/Optimum/Early End | Time [S] | Iterations |
|---|---|---|---|
| $10 \times 6$ | 7720/7720/no | 0.75 | 31,535 |
| $11 \times 5$ | 7038/7038/no | 0.13 | 243 |
| $12 \times 5$ | 7312/7312/no | 0.42 | 13,095 |
| $13 \times 4$ | 7166/7166/no | 0.20 | 650 |
| $14 \times 4$ | 8003/8003/no | 0.13 | 262 |
| $20 \times 10$ | 1566/1566/no | 164.45 | 2,619,405 |
| $30 \times 10$ | 2120/2093/t-l-e | 1000.02 | 6,398,821 |
| $30 \times 15$ | 2692/2513/t-l-e | 1000.02 | 4,886,367 |
| $50 \times 10$ | 3190/3045/t-l-e | 1000.03 | 3,164,599 |
| $75 \times 20$ | 5372/in [4890, 4951]/t-l-e | 1000.03 | 2,145,971 |

To do this, we will need a model that builds an appropriate schedule for the permutation. The genetic algorithm will then select a promising part of the search space of permutations in which a good approximation of the optimum can be found in a reasonable amount of time.

If we have processing times $p_{ij}$ for job $i$ on machine $j$, and a job permutation $J_1, J_2, \ldots, J_n$, then we can calculate the completion times $C_{J_i, j}$ as follows:

$$C_{J_1,1} = p_{J_1,1} \tag{90}$$
$$\forall i \in J - \{1\} : C_{J_i,1} = C_{J_{i-1},1} + p_{J_i,1} \tag{91}$$
$$\forall k \in M - \{1\} : C_{J_1,k} = C_{J_1,k-1} + p_{J_1,k} \tag{92}$$
$$(\forall i \in J - \{1\})(\forall k \in M - \{1\}) : C_{J_i,k} = \max\{C_{J_{i-1},k}, C_{J_i,k-1}\} + p_{J_i,k} \tag{93}$$
$$C_{\max} = C_{J_n,m} \tag{94}$$

As the genetic algorithms [79] are well known, we only summarise parameter settings and describe only the problem-specific operators in more detail.

The fitness function is inversely proportional to the makespan, the smaller the makespan, the higher the value of the fitness function.

The number of individuals in the population was set to 50 and the number of iterations to $10n^2$. The initial population was generated randomly, and the parents for the crossover operation were determined by binary tournament selection.

As to the *crossover* operation, we cannot use the traditional two-point crossover, because it would lead to infeasible solutions. If we change the middle parts of the parent chromosomes $P_1$ and $P_2$ in Figure 2, we will obtain offspring (10,5,2,6,10,4,1,6,3,1) and (5,8,4,7,8,9,2,6,3,1) that correspond to no permutations, because some jobs are duplicated or omitted. We used what is called *crossover in a partially mapped representation* where the genes in the middle part of one chromosome are ordered in its offspring by their occurrence in the second parent chromosome.

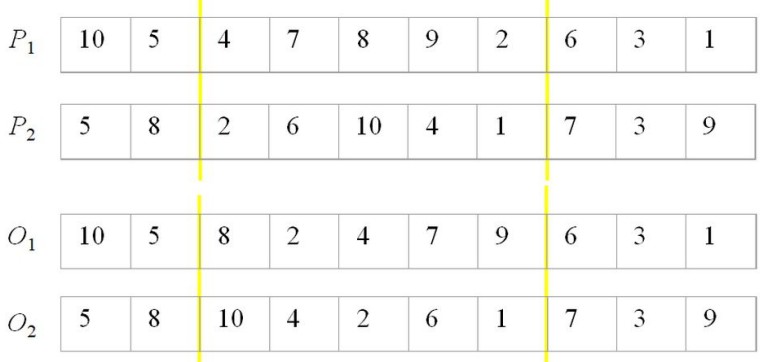

**Figure 2.** Modified two-point crossover.

From the possible mutation operations, we have selected the *shift mutation*, which removes a value at one position and puts it at another position), see Figure 3.

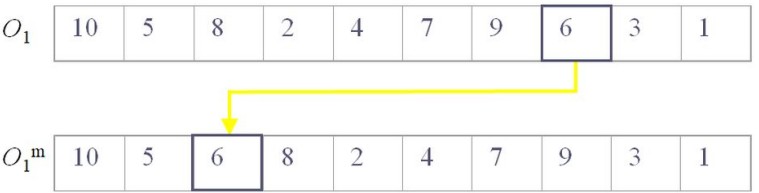

**Figure 3.** Shift mutation.

The offspring of the parents replaced two randomly selected individuals with below-average fitness function values.

With the above parameter settings, the results for the last 4 instances in Table 1 were obtained as shown in Table 2. The average values from 30 runs are presented here, as well as the best values obtained from them, which for these large instances are better than the values obtained from GAMS when the 1000 s timeout expires. All these results were achieved in less than 10 s because of the small number of iterations of the genetic algorithm.

**Table 2.** GA computational results—average and the best result from 30 runs, optimum.

| Benchmark | Average Result/the Best Result | Optimum |
|---|---|---|
| $30 \times 10$ | 2126/2099 | 2093 |
| $30 \times 15$ | 2570/2525 | 2513 |
| $50 \times 10$ | 3132/3090 | 3045 |
| $75 \times 20$ | 5261/5203 | between 4890 and 4951 |

### 6.2. TSP Implementation in GAMS

In describing the source code in GAMS and verifying its computational abilities, we use three benchmarks from the TSPlib library [83] with 24, 52, and 100 cities, or positions in the map given by coordinates.

The following code is written for the `gr24.tsp` benchmark. Since the adjacency matrix is symmetric, only the data of the lower triangular matrix are entered with the remaining data calculated. The EQUATIONS section is a rewrite of the TSP model and its Equations (5)–(8). The $x_{ij}$ binary domain, corresponding to Equation (9), is given by the declaration that precedes this section.

```
$TITLE Travelling Salesman Problem
OPTION ITERLIM=10000000;
OPTION OPTCR=0;

SETS
  I /1*24/;
```

```
    ALIAS (I,J);

PARAMETERS
  N;
  N=CARD(I);

TABLE C(I,J) adjacency matrix
         1    2    3    4    5    6    7    8    9   10   11   12   13   14   15   16   17   18   19   20   21   22   23   24
 1    0
 2   257    0
 3    87  196    0
 4    91  228  158    0
 5   150  112   96  120    0
 6    80  196   88   77   63    0
 7   130  167   59  101   56   25    0
 8   134  154   63  105   34   29   22    0
 9   243  209  286  159  190  216  229  225    0
10   185   86  124  156   40  124   95   82  207    0
11   214  223   49  185  123  115   86   90  313  151    0
12    70  191  121   27   83   47   64   68  173  119  148    0
13   272  180  315  188  193  245  258  228   29  159  342  209    0
14   219   83  172  149   79  139  134  112  126   62  199  153   97    0
15   293   50  232  264  148  232  203  190  248  122  259  227  219  134    0
16    54  219   92   82  119   31   43   58  238  147   84   53  267  170  255    0
17   211   74   81  182  105  150  121  108  310   37  160  145  196   99  125  173    0
18   290  139   98  261  144  176  164  136  389  116  147  224  275  178  154  190   79    0
19   268   53  138  239  123  207  178  165  367   86  187  202  227  130   68  230   57   86    0
20   261   43  200  232   98  200  171  131  166   90  227  195  137   69   82  223   90  176   90    0
21   175  128   76  146   32   76   47   30  222   56  103  109  225  104  164   99   57  112  114  134    0
22   250   99   89  221  105  189  160  147  349   76  138  184  235  138  114  212   39   40   46  136   96    0
23   192  228  235  108  119  165  178  154   71  136  262  110   74   96  264  187  182  261  239  165  151  221    0
24   121  142   99   84   35   29   42   36  220   70  126   55  249  104  178   60   96  175  153  146   47  135  169    0;

SET C2(I,J);
C2(I,J)$(NOT SAMEAS(I,J)) = yes;
C(C2(I,J)) = MAX(C(I,J),C(J,I));

VARIABLES
  X(I,J)
  delta(I)
  Z;
BINARY VARIABLE X(I,J);

EQUATIONS
  EQ1(J)   each city is entered exactly once
  EQ2(I)   each city is left exactly once
  EQ3(I,J) subtour elimination constraints
  EQ4      objective function;
  EQ1(J) .. SUM(I,X(I,J)$(ORD(I) NE ORD(J))) =E= 1;
  EQ2(I) .. SUM(J,X(I,J)$(ORD(I) NE ORD(J))) =E= 1;
  EQ3(I,J)$((ORD(I) GE 2) AND (ORD(J) GE 2) AND (ORD(I) NE ORD(J)))
        .. delta(I)-delta(J)+N*X(I,J) =L= N-1;
  EQ4    .. Z =E= SUM((I,J),C(I,J)*X(I,J));

MODEL TSP/ALL/;
SOLVE TSP USING MIP MINIMIZING Z;
DISPLAY X.L, Z.L;
```

The total length of the route is 1272, the decision variables $x_{i,j}$ have a value of 1 in the following order: $x_{1,16}$, $x_{16,11}$, $x_{11,3}$, $x_{3,7}$, $x_{7,6}$, $x_{6,24}$, $x_{24,8}$, $x_{8,21}$, $x_{21,5}$, $x_{5,10}$, $x_{10,17}$, $x_{17,22}$, $x_{22,18}$, $x_{18,19}$, $x_{19,15}$, $x_{15,2}$, $x_{2,20}$, $x_{20,14}$, $x_{14,13}$, $x_{13,9}$, $x_{9,23}$, $x_{23,4}$, $x_{4,12}$, $x_{12,1}$, and, thus, the circuitous route passes through cities 1, 16, 11, 3, 7, 6, 24, 8, 21, 5, 10, 17, 22, 18, 19, 15, 2, 20, 14, 13, 9, 23, 4, 12, 1, which is in agreement with the published result for the gr24 benchmark. The calculation time was 0.36 s.

The data for the berlin52.tsp benchmark are entered differently, namely as a matrix with 52 rows and 2 columns, where the 1st column is the value of the *x*-coordinate and the 2nd column is the value of the *y*-coordinate. The adjacency matrix in this case is obtained by calculating the Euclidean distances between all pairs of positions. This part of the code takes the following form, the rest is the same as in the previous code.

```
SETS
  I /1*52/;
  ALIAS (I,J);

PARAMETERS
  N;
```

```
   N=CARD(I);

TABLE XY(I,*)
        1        2
1  565.0    575.0
2   25.0    185.0
3  345.0    750.0
4  945.0    685.0
5  845.0    655.0
6  880.0    660.0
7   25.0    230.0
8  525.0   1000.0
9  580.0   1175.0
10 650.0   1130.0
11 1605.0   620.0
12 1220.0   580.0
13 1465.0   200.0
14 1530.0     5.0
15  845.0   680.0
16  725.0   370.0
17  145.0   665.0
18  415.0   635.0
19  510.0   875.0
20  560.0   365.0
21  300.0   465.0
22  520.0   585.0
23  480.0   415.0
24  835.0   625.0
25  975.0   580.0
26 1215.0   245.0
27 1320.0   315.0
28 1250.0   400.0
29  660.0   180.0
30  410.0   250.0
31  420.0   555.0
32  575.0   665.0
33 1150.0  1160.0
34  700.0   580.0
35  685.0   595.0
36  685.0   610.0
37  770.0   610.0
38  795.0   645.0
39  720.0   635.0
40  760.0   650.0
41  475.0   960.0
42   95.0   260.0
43  875.0   920.0
44  700.0   500.0
45  555.0   815.0
46  830.0   485.0
47 1170.0    65.0
48  830.0   610.0
49  605.0   625.0
50  595.0   360.0
51 1340.0   725.0
52 1740.0   245.0;

PARAMETERS C(I,J);

SET C2(I,J);
C2(I,J)$(NOT SAMEAS(I,J)) = yes;
C(C2(I,J)) = ROUND(SQRT(SQR(XY(I,'1')-XY(J,'1'))+SQR(XY(I,'2')-XY(J,'2'))));
```

The total length of the route is 7542, the calculation time was 1.45 s and the circuitous route passes through positions 1, 49, 32, 45, 19, 41, 8, 9, 10, 43, 33, 51, 11, 52, 14, 13, 47, 26, 27, 28, 12, 25, 4, 6, 15, 5, 24, 48, 38, 37, 40, 39, 36, 35, 34, 44, 46, 16, 29, 50, 20, 23, 30, 2, 7, 42, 21, 17, 3, 18, 31, 22, 1. The optimal route can be seen in Figure 4.

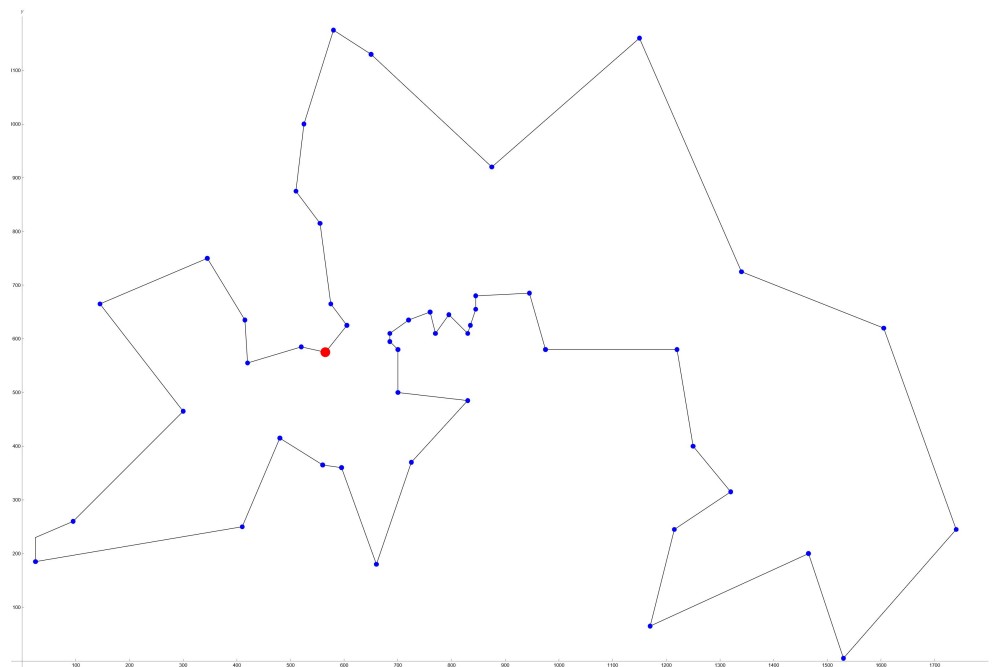

**Figure 4.** The optimal route for the berlin52.tsp benchmark.

Finally, GAMS for the `kroA100.tsp` benchmark stopped the computation at 1000.02 s by exceeding the time limit, but the intermediate result of the path length 21282 and its traversal through positions 1, 47, 93, 28, 67, 58, 61, 51, 87, 25, 81, 69, 64, 40, 54, 2, 44, 50, 73, 68, 85, 82, 95, 13, 76, 33, 37, 5, 52, 78, 96, 39, 30, 48, 100, 41, 71, 14, 3, 43, 46, 29, 34, 83, 55, 7, 9, 57, 20, 12, 27, 86, 35, 62, 60, 77, 23, 98, 91, 45, 32, 11, 15, 17, 59, 74, 21, 72, 10, 84, 36, 99, 38, 24, 18, 79, 53, 88, 16, 94, 22, 70, 66, 26, 65, 4, 97, 56, 80, 31, 89, 42, 8, 92, 75, 19, 90, 49, 6, 63, 1 corresponds to the known optimal solution for this benchmark. The optimal route is in Figure 5.

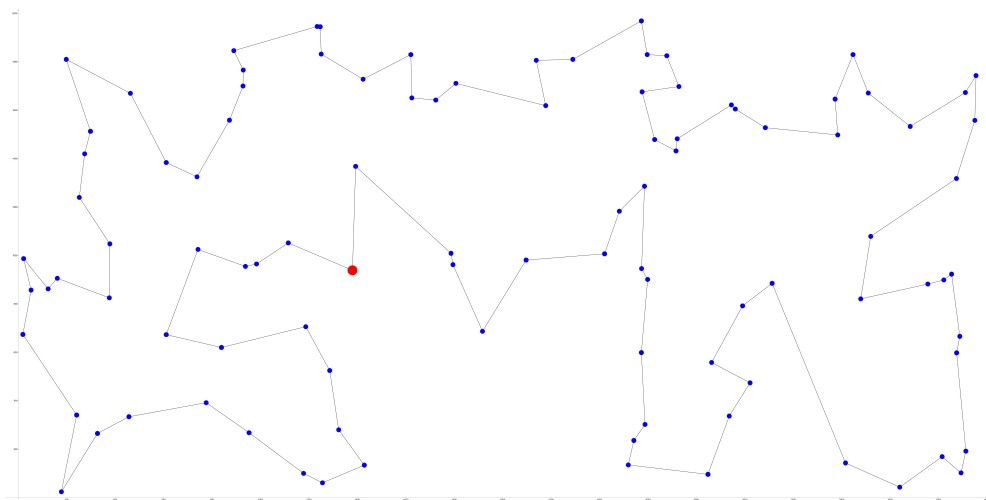

**Figure 5.** The optimal route for the kroA100.tsp benchmark.

For instances with more than 100 positions, it would be necessary to search for an approximation of the optimum using one of the heuristic methods.

One of the first was the use of the so-called *Lin-2-Opt change operator* [8], see Figure 6. Here, two elements are added to the permutation of $n$ cities to visit (into positions 0 and $n + 1$), and then the starting city is assigned to those positions to simulate a cyclic tour. Two 'edges' (pairs of neighbouring elements in permutation) are randomly chosen (($p_1$, $p_2$) and

$(q_1, q_2)$ say), the inner elements $p_2$, $q_1$ are swapped and the elements between $p_2$ and $q_1$ are reversed.

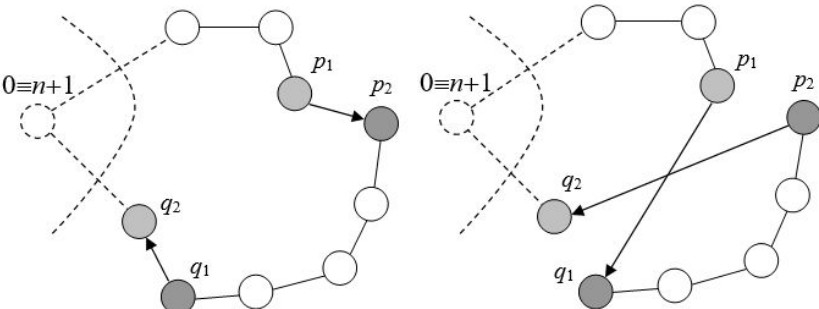

**Figure 6.** Lin-2-Opt change neighbourhood operation,

Positions 0 and $n + 1$ with the fixed value of the starting city can also be used to expand the individuals in the population of the genetic algorithm and then apply the operations presented for the PFSSP.

However, we no longer investigate this for extremely large instances of the TSP problem because heuristics do not guarantee finding an optimal solution, which often is not even known here, and the aim was to find bounds for which we still obtain the precise solution in reasonable time using a 'normal' computer. In the case of GAMS, this bound is an instance with 100 cities.

### 6.3. Data, Changes in Time, Uncertainty

Data from OR-Library and TSPLIB are related to a specific point in time, in reality they may change over time or may not be completely known.

A more general case of the Travelling Salesman Problem is the *Canadian Traveller Problem* (CTP) [84,85]. Here, the distance matrix may change over time due to the occurrence of events that make some parts of the route inaccessible so that an adaptive strategy must be found. These events are random in nature, which corresponds to the problems of robot navigation in environments where the distribution of obstacles is only discovered as the robot moves through the environment; moreover, the obstacles may move, and thus the locations of potential collisions change dynamically.

In transport tasks, the values of some parameters can change over time, the fuel price is not constant, and the vehicle consumption can only be estimated because it can change according to the traffic situation and the season, which will affect, e.g., the calculation of the objective function (69). Similarly, in the crop problem, we can only estimate crop yields.

In location-based tasks, the problem may arise of adding another center to an existing network of centers to improve the coverage of an area. An example might be an expansion of the existing supermarket network of a chain store. Here it is suggested to use one of the properties of the Voronoi diagram [86,87], a data structure known from computational geometry: Assume a Voronoi diagram with its sites represented by the current centers. The point $q$ is the vertex of the Voronoi diagram if and only if the *largest empty circle* $C(q)$ contains three (or more in a degenerate case) sites on its boundary and none inside. Among these circles, we determine the one with the largest diameter, and its center is then the optimal position for the location of the new center, see Figure 7.

In fact, the calculated position may not be available, the cost of building here may be too high, thus a suitable nearby location must be found, or the center of one of the other empty circles must be chosen in descending order of diameters.

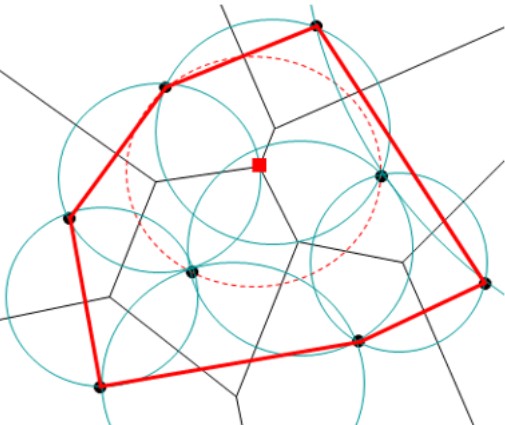

**Figure 7.** Finding a new location using the largest empty circles.

Another issue is the amount of inventory in logistics operations. The stock changes over time according to demand and needs to be replenished accordingly. We speak about *inventory management* and the resulting sustainability [88–91]. However, demands are stochastic in nature and, in addition, inventory management must take into account the cost of maintaining inventory and losses from premature depletion of inventory for undelivered goods.

In [92], a new mathematical model is derived, the properties of the profit function are proved, and the profitability in a two-channel production system considering carbon emissions and green technology is numerically verified on specific data.

While artificial neural networks (ANN) have very little application in combinatorial optimization, their main use is in cluster analysis, pattern recognition, image processing and prediction, in [93] the authors present an efficient implementation of ANNs in an inventory management model under uncertainty and inflation.

In [94] the unreliability of the supply chain and methods to eliminate this unreliability are explored, and the required mathematical equations are derived and verified by numerical experiments, including sensitivity analysis.

However, all these aspects are beyond the scope of this paper and can be the subject of separate texts as also evidenced by the papers mentioned.

## 7. Conclusions

This paper studies the assignment problem and its modifications with logistics applications, in routing, distribution, and scheduling tasks. Its first contribution is the correlation of the problem models, which are often distant in nature and time complexity.

It has also shown how the described models can be directly transferred to the GAMS environment. NP-hard Permutation Flow Shop Scheduling Problem (PFSSP) and the Travelling Salesman Problem are used to show that the optimal solution can be determined in the available time of a few minutes for instances with 20 jobs on 10 machines in the case of PFSSP, and for 100 cities in the case of TSP.

Previously, these boundaries were inaccessible with mixed integer programming solvers, but with the new version of GAMS, they have been significantly extended. This of course means first to build the appropriate model (and this is not always a simple matter, as the informal derivation of the PFSSP model in Section 5 showed) and then, for instance, for benchmark libraries (e.g., OR-Library or TSPLIB), to search individually for the appropriate bounds. The findings from PFSSP and TSP are not isolated examples of the successful application of GAMS in solving large instances of optimization versions of NP-complete/NP-hard problems. We have already validated it in [13] in solving the covering problem with matrices of hundreds of thousands of elements, and more recently in solving the Steiner problem in graphs in [95], where first using the terminology of network flows

a mixed integer programming model was derived, then modified for GAMS, and finally exact results for a representative class of benchmarks from OR-Library were obtained.

Another goal of this paper was to introduce code generation in GAMS on non-trivial tasks because in the manuals [76,77] we can find only a description of individual elements of this tool, but not the codes of complete task models. In MATLAB, running the computation of an optimization program means writing just a single command (`intlinprog` or `linprog` with the appropriate parameters). Similarly, when solving differential equations, e.g., to calculate the differential equation $y'(x) = 4xy + x^3$ with initial condition $y(4) = 2$, it is enough to enter `dsolve('Dy=4*x*y+x^3','y(4)=2','x')`. In MATHEMATICA, too, to obtain the impulse function of the system described by a differential equation, it is enough to rewrite it in the form of a Laplace transfer and use a single command `InverseLaplaceTransform`. In contrast, the code notation in GAMS is similar to code in programming languages with the definition of constants, the declaration of variables, and the body of the program. Again, there are assignment statements, conditional statements, and loop statements. For example, the binary values of the reachability matrix **A** from the distance matrix **D** and the defined reachable distance threshold $D_{max}$ are determined in GAMS as follows:

```
LOOP(I,
      LOOP(J,
            IF (D(I,J) <= Dmax,
                A(I,J)=1;
              ELSE
                A(I,J)=0;
              );
          );
    );
```

The only disadvantage of GAMS is that it has no graphical tools, and the results of the calculations are only in text form. This requires exporting them to a suitable program and postprocessing. In [13] we used MATLAB, here Figures 4 and 5 are generated in the MATHEMATICA environment.

Only where for extremely large instances of problems of exponential complexity we cannot obtain an exact solution using GAMS in a reasonable amount of time (e.g., no more than in tens of minutes), do we use one of the many heuristic methods. Given the No Free Lunch Theorem [74,75], none of them can be recommended as the best in the general case, since finding the optimal solution is not guaranteed and the result is always an approximation of the optimum, so our modification of the genetic algorithm, implemented in Java and described in more detail in [79], can be used without loss of generality.

One-point heuristics (hill climbing, tabu search, simulated annealing) in solving problems where in each iteration the neighborhood operation often generates tens of infeasible solutions and it is necessary to use a repair operator for them (here it concerns the coverage problem), and slow down the computation considerably, so in these cases we prefer, e.g., a genetic algorithm that generates only two new solutions in each iteration.

The model in Section 4.4 is original with another possible modification proposed at the end. Its model has already been built and verified on smaller-scale instances so far and will be investigated in more complex cases.

In future research, we expect to focus on the Quadratic Assignment Problem, the Vehicle Routing Problem and its solvability using GAMS, and applications in agriculture with consideration of data uncertainty using probabilistic models or fuzzy modeling, since yields can only be estimated. Although the Quadratic Assignment Problem has a non-linear objective function with quadratic terms, it can be converted to a mixed integer programming problem using Lawler's linearization [7] and the MIP solver of GAMS can be used again.

**Funding:** This research received no external funding.

**Data Availability Statement:** Not applicable.

**Conflicts of Interest:** The authors declare no conflict of interest.

## Abbreviations

The following abbreviations are used in this manuscript:

| | |
|---|---|
| AP | Assignment Problem |
| TSP | Travelling Salesman Problem |
| VRP | Vehicle Routing Problem |
| PFSSP | Permutation Flow Shop Scheduling Problem |
| GAMS | General Algebraic Modelling System |
| GA | Genetic Algorithm |
| ANN | Artificial Neural Network |

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
