# Peer review of "The Assignment Problem and Its Relation to Logistics Problems"

_algorithms, doi:10.3390/a15100377_

Round 1
Reviewer 1 Report (Previous Reviewer 2)
Sorry, I would have responded sooner, but I am just recovering from Covid.
Okay, one more time I have clicked "reconsider" but I don't think the revision I need to see is major.
(1) You say: "And getting such a result for a TSP instance with 100 cities seems to me a surprising result." That is part of my problem. I suspect that GAMS is doing something clever that we are missing and that the scaling laws are not what you think they are. You also say "The goal was to find the boundary where GAMS still finds the exact solution under ’normal conditions’ and in a reasonable amount of time." You can't really find a boundary with one data point. What I *need* to see is that you've tried with larger resources and found a practical sort of boundary for your problem. Without this, I can't help but feel you are missing an essential part of the science.
(2) Please update your conclusion to match what you wrote in response to my question about what the novel contributions of the paper are. You should also say something about the genetic algorithms that you employed/developed.
Author Response
Please see the attachment.

Reviewer 2 Report (Previous Reviewer 3)
This study "The assignment problem and its relation to logistics problems" is based on programming and methodology. Update the following comments.
1. What are the data sources? Validate input data for computational results.
2. Provide a comparative study with existing literature to establish the novelty of the study. Make a subsection 6.3 for comparative analysis "numerically" and provide managerial insights of this study in another subsection 6.4.
3. Explain the conclusions section more based on the insights of the study.
4. For reference of comparative study and others, study the followings for literature review.
a. A sustainable managerial decision-making problem for a substitutable product in a dual channel under carbon tax policy
b. Application of the artificial neural network with multithreading within an inventory model under uncertainty and inflation.
c. Economic and environmental assessment of an unreliable supply chain management
Round 2
Reviewer 1 Report (Previous Reviewer 2)
The manuscript has improved a lot over the revisions. I think it is ready, but I think the authors should continue to investigate the effects of using more powerful computers.
Reviewer 2 Report (Previous Reviewer 3)
Thank you for the revision.
This manuscript is a resubmission of an earlier submission. The following is a list of the peer review reports and author responses from that submission.
Round 1
Reviewer 1 Report
The paper presents a theoretical study aiming to examine the assignment problem and its modifications, with logistics applications, in routing, distribution and scheduling tasks. It has also been shown how the described models can be directly transferred to the GAMS environment, and on benchmarks for NP-hard Permutation Flow Shop Scheduling Problem (PFSSP) and the Travelling Salesman Problem.
The assignment problem is a fundamental combinatorial optimization problem. The problem instance has a number of agents and a number of tasks. Any agent can be assigned to perform any task, incurring some cost that may vary depending on the agent-task assignment. It is required to perform as many tasks as possible by assigning at most one agent to each task and at most one task to each agent, in such a way that the total cost of the assignment is minimized. Alternatively, the assignment problem consists of finding, in a weighted bipartite graph, a matching of a given size, in which the sum of weights of the edges is minimum.
The paper has shown that the optimal solution can be determined in the available time of a few minutes for instances with jobs on 10 machines in the case of PFSSP and for 100 cities in the case of TSP. Instead of following the traditional approach based on the use of approximate or stochastic heuristic methods, the paper uses directly mixed integer programming models in the GAMS environment, which is capable of solving much larger instances than in the past and does not require complex parameter settings and statistical evaluation of the results.
The discussion and explanations provided by the authors about the findings are quite analytical and sufficient, stand in the appropriate level and do not extend into useless details. Quality of presentation is quite good and the reader of the paper obtains easily and quickly an overview of the findings of the investigation performed. The subject of this investigation is quite important in the area of novel simulation and prediction methodologies and offers several important conclusions. Therefore the paper is suggested for publication.
Reviewer 2 Report
I hate to give a paper such low scores. The first section, describing how the TSP and other problems are related to each other was interesting. I'm not sure it counts as a novel result, but it was interesting.
The second part of the problem had some interesting content with regard to the genetic algorithm, but the remainder seemed to be "we ran it through GAMS and GAMS solved everything." There was no discussion of how GAMS solved the problem (yes, there was source code, but this is no substitute for a discussion, especially since the code seemed to do nothing but declare the problem).
There were some clarity issues as well.
I assume "za podmínek" means "subject to?" Regardless, please translate.
I'm a little confused by section 6. First you say that you select TSP and PFSSP, then just before section 6.1 you say you start with PFFP since it's more complex than TSP. I don't know what the PFFP problem is, but I am guessing that it is the same as PFSSP. I'm also guessing that PFFF is another variant on PFFP and PFSSP? Regardless, neither PFFF nor PFFP are defined in the text.
In this section a laptop with 2.4GHz is mentioned, but from the text that follows it appears that the amount of memory the machine has is the most important factor. I also wonder why only a laptop is used instead of larger more capable machine.
Section 6.1 is labeled "PFSSP Computational Results." This makes sense if, as I suspect, PFFP is the same as PFSSP.
Reviewer 3 Report
In the article "The Assignment Problem and Its Relation to Logistics Problems", the authors want to explain different logistics system scenarios but fail to provide novelty.
1. Literature survey is missing.
2. There are a lot of problems like crop problems, transportation problems in SCM, location problems, etc. But the discussion is about only one problem, the scheduling problem. This is too confusing. What is the purpose of stating the basics without stating the novelty of the research?
3. There is no comparison with existing literature to prove the establishment and novelty of this study.
4. It feels like the study is almost three years old. Then the contributions need to be established but the proof of the novelty is missing.
Round 2
Reviewer 2 Report
I am still struggling with a few issues. First, why limit yourself to a "normal" computer? Are the users of GAMS expected to have such limited access to computers? It seems to me (from the nature of the problems you discuss) that the normal users would be companies who would readily invest in a bigger machine. Alternatively, it seems odd that an academic (who would have access to campus machines) would not take the trouble to gain access and use them. What is the reason for the limitation? If there is no reason, then please try running on a bigger machine.
The 2nd and bigger problem, though, is one I tried to raise before and I think you did not understand what I was asking. What is your contribution? It seems to be this: "It has also shown how the described 671 models can be directly transferred to the GAMS environment." Was there some special challenge in transferring these problems to GAMS? Phrasing this another way, if I plug a known differential equation into Mathematica and it gives me a solution, is this worth a paper? I would say not. Likewise, why is plugging a standard problem into GAMS and getting an answer back a contribution? Please clarify.
I'm clicking "reconsider after major revision" although I am focusing on the "reconsider" part. I'm hoping that you can address the above two questions without a large modification to the text.
Author Response
Dear reviewer,
let me respond to your last comments and thank you for them:
1.
“I am still struggling with a few issues. First, why limit yourself to a "normal" computer? Are the users of GAMS expected to have such limited access to computers? It seems to me (from the nature of the problems you discuss) that the normal users would be companies who would readily invest in a bigger machine. Alternatively, it seems odd that an academic (who would have access to campus machines) would not take the trouble to gain access and use them. What is the reason for the limitation? If there is no reason, then please try running on a bigger machine.”
-->
Of course, with a computer of better technical parameters for the same time limit we get results for larger instances of the problem, but that's not the most important thing, e.g., the time complexities of algorithms are given in number of operations and not in computation time.
The goal was to find the boundary where GAMS still finds the exact solution under “normal conditions“ and in a reasonable amount of time. And getting such a result for a TSP instance with 100 cities seems to me a surprising result.
Beyond this boundary, we use heuristics. We have already presented a new modification of the genetic algorithm in [77] Seda, P.; Mark, M.; Su, K.W.; Seda, M.; Hosek, J.; Leu, J.: The Minimization of Public Facilities with Enhanced Genetic Algorithms. IEEE Access 2019, 7, 9395–9405 (in other words in solving large instances for a minimisation of network covering services in a threshold distance). This heuristic (in our own Java implementation) could perhaps be used in a separate paper only on TSP and compared with the results of other approaches, but the focus of this paper is a survey.
2.
“The 2nd and bigger problem, though, is one I tried to raise before and I think you did not understand what I was asking. What is your contribution? It seems to be this: "It has also shown how the described models can be directly transferred to the GAMS environment." Was there some special challenge in transferring these problems to GAMS? Phrasing this another way, if I plug a known differential equation into Mathematica and it gives me a solution, is this worth a paper? I would say not. Likewise, why is plugging a standard problem into GAMS and getting an answer back a contribution? Please clarify.”
I'm clicking "reconsider after major revision" although I am focusing on the "reconsider" part. I'm hoping that you can address the above two questions without a large modification to the text.“
-->
In my opinion, the paper is an unconventional survey with the addition of solvability in an environment that is also not common as, e. g., Optimisation Toolbox in MATLAB, something like the "first chapter" of the special issue. I also think that not all equations can be found in Operation Research textbooks, e.g., equation (49) and its derivation, an informal derivation of a model of the permutation scheduling problem.
I also think that not all equations/inequalities can be found in operations research textbooks, e.g., equation (49) and its derivation and extended model in Section 4.4, or the informal derivation of a MIP model of the PFSSP.
"MATHEMATICA, differential equations, …"
I also work with this software, Figures 4 and 5 are generated in it, because GAMS does not have a graphical tool, I also use it in control theory to draw step response and impulse characteristics of systems described by differential equations or Laplace transfers. Admirably, it also outputs the analytical solutions of these responses alongside the graph, but from a user point of view, only the InverseLaplaceTransform and Plot commands are sufficient to call them.
Similarly, in MATLAB, one command is enough to run the differential equation solver, e.g. solution=dsolve('Dy=4*x*y+x^3','y(4)=2','x').
In this respect, building code for GAMS is much more complex and the examples chosen may be helpful to a potential reader.
Reviewer 3 Report
Thank you for your effort. Good luck!
Author Response
Dear reviewer,
apparently you have definitely rejected the text because you don't provide further comments, yet I will add what I wrote to the second reviewer who admitted another reply, because maybe it is also related to your comments from the first review.
-->
Of course, with a computer of better technical parameters for the same time limit we get results for larger instances of the problem, but that's not the most important thing, e.g., the time complexities of algorithms are given in number of operations and not in computation time.
The goal was to find the boundary where GAMS still finds the exact solution under “normal conditions“ and in a reasonable amount of time. And getting such a result for a TSP instance with 100 cities seems to me a surprising result.
Beyond this boundary, we use heuristics. We have already presented a new modification of the genetic algorithm in [77] Seda, P.; Mark, M.; Su, K.W.; Seda, M.; Hosek, J.; Leu, J. The Minimization of Public Facilities with Enhanced Genetic Algorithms. IEEE Access 2019, 7, 9395–9405 (in other words in solving large instances for a minimisation of network covering services in a threshold distance). This heuristic (in our own Java implementation) could perhaps be used in a separate paper only on TSP and compared with the results of other approaches, but the focus of this paper is a survey.
-->
In my opinion, the paper is an unconventional survey with the addition of solvability in an environment that is also not common as, e. g., Optimisation Toolbox in MATLAB, something like the "first chapter" of the special issue. I also think that not all equations can be found in Operation Research textbooks, e.g., equation (49) and its derivation, an informal derivation of a model of the permutation scheduling problem.
I also think that not all equations/inequalities can be found in operations research textbooks, e.g., equation (49) and its derivation and extended model in Section 4.4, or the informal derivation of a MIP model of the PFSSP.
"MATHEMATICA, differential equations, …"
I also work with this software, Figures 4 and 5 are generated in it, because GAMS does not have a graphical tool, I also use it in control theory to draw step response and impulse characteristics of systems described by differential equations or Laplace transfers. Admirably, it also outputs the analytical solutions of these responses alongside the graph, but from a user point of view, only the InverseLaplaceTransform and Plot commands are sufficient to call them.
Similarly, in MATLAB, one command is enough to run the differential equation solver, e.g. solution=dsolve('Dy=4*x*y+x^3','y(4)=2','x').
In this respect, building code for GAMS is much more complex and the examples chosen may be helpful to the potential reader.